# Examining the Relationship between Pro-Environmental Attitudes, Self-Determination, and Sustained Intention in Eco-Friendly Sports Participation: A Study on Plogging Participants

**Jongho Kim** [1] , **Sujin Kim** [2] **and Jinwook Chung** [2,*]

1   Department of Physical Education, Seoul National University, Seoul 08826, Republic of Korea; kikara77@snu.ac.kr
2   Department of Sport Culture, Dongguk University, Seoul 04620, Republic of Korea; 03sujin27@gmail.com
*   Correspondence: cjw826@dongguk.edu

**Abstract:** In response to rising environmental concerns and the increase in eco-friendly sports activities, this study investigated the determinants of sustained intention to participate in plogging, a combination of jogging and litter collection. A total of 288 randomly assigned plogging participants were surveyed to discern the effects of autonomy, competence, and relatedness experiences on sustained plogging intentions as suggested by self-determination theory. The study also examined the moderating role of eco-friendly attitudes. The analysis, executed using multi-group structural equation modeling, revealed that while autonomy and competence did not significantly influence extrinsic motivation, relatedness emerged as the most influential factor. This suggests that plogging primarily serves as a prosocial behavior, enhancing relationships, rather than a means to increase physical competence. The values derived from plogging and the intention to continue varied based on the participants' eco-friendly attitudes. The authors conclude that voluntary participation and socialization are the core values of plogging and understanding these can promote healthier and more sustainable behaviors.

**Keywords:** plogging; exercise participation; self-determination theory

## 1. Introduction

Global environmental issues such as global warming, food and water scarcity, and energy depletion have been highlighted as critical challenges that need to be addressed [1]. In this context, the growing awareness among many members of society regarding the impact of their actions on the planet has resulted in the emergence of significant societal agendas, exemplified by terms such as 'environment', 'eco-friendly', 'sustainability', and 'green' [2]. For instance, the United Nations announced the 'Sustainable Development Goals', urging environmentally friendly actions (United Nations, 2015). Many large corporations appeal to consumers through green marketing strategies using environmental, social, and governance (ESG) management [3,4]. From this perspective, mature consumers are demonstrating a tendency towards 'pro-environmental behavior', considering both environmental protection and consumption behavior [5]. This phenomenon is spreading across various fields, adjusting human lifestyles in diverse ways.

The tendency towards eco-friendly sports is also evident in sports consumption and participation behaviors [6]. Notable examples include the organization of eco-friendly sports events, the promotion of eco-friendly ideologies through sports, and the development and expansion of eco-friendly sports initiatives [6]. One such eco-friendly sport is plogging, which originated in Sweden and involves combining jogging with the act of picking up discarded litter. Plogging serves as a volunteer activity for environmental

protection while also promoting individual health and well-being [7–9]. The engagement of individuals in these activities demonstrates their active participation in environmental preservation [7–9]. Moreover, these eco-friendly sports contribute to personal life satisfaction, the formation of self-identity, and overall happiness, as they simultaneously prioritize individual health and well-being along with environmental protection campaigns [10]. Previous research has highlighted the positive impact of participating in eco-friendly sports on personal life satisfaction, competence development, and physical fitness [7–9,11].

However, encouraging the intention for sustained participation in eco-friendly sports behavior, which is a serious leisure activity, is challenging as it cannot easily be carried out through regulatory measures or economic incentives [12]. For the positive social spread of eco-friendly sports and the personal life satisfaction and well-being of its participants to operate smoothly, it is necessary to understand how to induce intention for sustained participation [13–15]. In general, previous studies analyzing various socio-psychological factors that influence the intention for sustained participation in sports or physical activities primarily suggest that it occurs through the satisfaction of basic psychological needs and the formation of motivation, as suggested by the 'Self-Determination Theory' [16–19]. However, eco-friendly sports are both physical activities for promoting physical health and volunteer activities for conserving nature, so it is necessary to understand eco-friendly behavior in addition to the theory related to sports participation. In particular, studies related to the intention for the sustained participation of volunteers commonly suggest the importance of voluntary participation motivation using the public service motivation theory and report that the formation of eco-friendly attitudes has a significant influence on the formation of voluntary participation motivation [12,20]. Therefore, to encourage and promote participation in eco-friendly sports, it is necessary to theoretically approach it from a convergent perspective by considering the variables suggested in the research analyzing the psychological motivation of sports participation and the variables explaining the motivation of eco-friendly behavior participation.

Therefore, the study objective is to explore and analyze strategies for enhancing the sustained participation intention of eco-friendly sports participants. This will be achieved by concurrently examining the variables associated with sports participation and eco-friendly behavior. Specifically, for this purpose, we analyzed how the factors and motivations involved in the intention for sustained participation, suggested by the self-determination theory, and eco-friendly attitudes that affect eco-friendly behavior, influence the continuous participation intention by targeting the participants of plogging, a representative eco-friendly sport. Through this, we aim to provide useful information for writing effective participation promotion strategies by understanding the fundamental psychological processes of eco-friendly sports participants, which can help in both aspects of personal well-being and environmental protection. This eco-friendly sports study is meaningful in that it applies existing theories to new research subjects, thereby broadening the base of related disciplines, and attempts a multidisciplinary theoretical exploration in that it considers the concepts of sports and volunteering simultaneously.

## 2. Theoretical Background and Research Hypotheses

### 2.1. Eco-Friendly Sports: Plogging

Eco-friendly sports encompass different forms of sports consumption, events, and participatory activities that prioritize environmental protection and sustainability [21–24]. Eco-friendly sports, which aim to respect and protect the natural environment, conserve resources, and promote environmental education and eco-friendly awareness, are currently being implemented in various ways. Examples include incorporating environmental values into existing sports events and combining eco-friendly activities with sports. Efforts were made at the 2020 Tokyo Olympics to build temporary facilities using recyclable materials and to form an eco-friendly power supply network by installing solar power generation systems and renewable energy power plants. A pre-event called 'spogomi', a competition to pick up as much trash as possible in a limited time, was also held,

and encouraged participation by demonstrating the positive social impact of eco-friendly sports [7]. Previous studies on eco-friendly sports have mainly focused on the production of sports goods [23,25], the importance of hosting eco-friendly sports events [22], and the importance of the ESG management concept in sports [26,27].

Meanwhile, as interest in participatory eco-friendly sports has increased, studies related to the meaning and uniqueness of participation experience and motivation in eco-friendly sports have also been actively conducted. Participatory eco-friendly sports are highlighted as important activities that not only increase physical activity and maintain healthy lifestyle habits, but also play a positive role in stress relief and enhancing life satisfaction at the personal level [9,28]. Participation in eco-friendly sports provides mental rest through communion with nature, and it is reported that positive emotions such as 'Helper's High' can be felt through altruistic volunteer activities [8,11]. Previous studies commonly reported that there was also an effect of strengthening bonds with others through the sharing of positive behaviors, and at the social level, it contributed to the improvement of the natural environment in the local community and the formation of social networks through the sharing of sports activities.

These eco-friendly sports are showing a trend of spreading mainly among the younger generation. A representative example is plogging, where many people around the world participate in picking up trash while jogging [7,11]. Previous studies conducted on plogging, where the attributes of an environmental protection campaign and exercise activities are combined, reported participation experiences and participation motivations in the special context of eco-friendly sports [8,29,30]. Raghavan, Panicker and Emmatty [11] suggested that plogging had a significant impact on aerobic exercise effects through jogging, as well as anaerobic lower body strength enhancement through squatting movements while picking up trash. Chae and Kim [9] reported that the participation efficacy of ploggers was influenced by the altruistic motivation of environmental protection, the motivation for self-development through sports activities, and the external motivation to enhance relationships with others, and that participating in a special form of eco-friendly sport that benefits others through their efforts played a positive role in personal values such as self-satisfaction. In addition, Yoon et al. [7] conducted in-depth interviews and reported that people who participated in plogging were aware of the seriousness of environmental pollution and had the will to execute environmental protection behaviors in their daily lives. Furthermore, they increased their intention for sustained participation through self-satisfaction and the fulfillment of the need for recognition through interaction with others. In conclusion, when synthesizing the results of these previous studies, it can be confirmed that plogging, which is a representative activity where the attributes of sports and volunteer activities are combined, acts as a producer of a sports culture that solves environmental problems through the act of sports itself, and participants acquire physical health and self-respect through the plogging experience. Many studies suggested the need for verification of social psychological variables to encourage special participation motivation and continuous participation in eco-friendly sports.

*2.2. Factors Influencing the Intention of Sustained Participation in Environmental Sports: Application of Self-Determination Theory*

As previously discussed, plogging, as a fitness activity and an environmental action, offers various benefits to individuals and society. Therefore, it is not only necessary for a large number of people to recognize and participate in the benefits of environmental sports, but it is also important to enhance the sustained participation intention of those who have experienced participation. This study aims to investigate the social and psychological factors necessary to promote and encourage sustained sports participation. For this purpose, we set up hypotheses about the factors influencing sustained participation intention and their relationships, focusing on the 'Self-Determination Theory' which has often been used in previous studies related to the intention of sustained sports or exercise participation.

Self-determination theory is a psychological theory proposed by Ryan and Deci [17,31] that explains human behavior motivation, and is often used to deeply analyze and explain the sustained participation motivation of exercise participants [32,33]. According to Ryan and Deci [17], humans have a fundamental tendency to pursue psychological need satisfaction, and intrapersonal motivation which supports this need satisfaction influences sustained participation intention, performance, pleasure, etc. Specifically, the self-determination theory presents a human's basic psychological needs as autonomy, competence, and relatedness. Furthermore, Vallerand [34] conveyed through an integrated model that the satisfaction or expectation of the three basic psychological needs stimulates the voluntary behavior will of intrinsic and extrinsic motivation that occurs according to rewards or punishments. In addition, this intrinsic and extrinsic motivation plays a positive role in sustained behavior intention. According to the results of past research mainly conducted in sports psychology, an individual's intrinsic motivation makes them recognize exercise behavior itself as a purpose, which influences sustained exercise performance behavior [35,36]. Furthermore, the results show that life sports participants increase their intrinsic and extrinsic motivation through the satisfaction of self-determination factors, thereby increasing their participation and sustained intention [37,38]. Given the context, this study considers eco-friendly sports as a form of exercise and physical activity. We then formulated our research hypotheses by applying the self-determination theory's framework, which suggests a relationship between the three fundamental psychological needs, intrinsic and extrinsic motivation, and the intention for sustained participation, to the behavioral intentions of eco-friendly sports participants.

Firstly, the need for autonomy refers to the basic human psychological desire to feel that the control of their actions lies within themselves, to consider themselves as the regulators of their own lives, and to act as they wish [17]. Participants in sports or physical activities feel autonomy when they choose to participate because they like it, forming a sense of responsibility for their own choices. This sense of responsibility acts as intrinsic motivation in itself, while also functioning as extrinsic motivation by forming a psychological reward and punishment system. Numerous studies have been modeling the role of motivation that mediates the relationship between the satisfaction of basic psychological needs and the intention to continue exercising and have been verifying it in various environments [19,32,33]. In particular, plogging is very important as a voluntary action because it is both healthy exercise and an altruistic environmental action. Therefore, the degree of satisfaction with the need for autonomy, as perceived through plogging participation, is predicted to have a positive impact on the intrinsic and extrinsic motivation of plogging participation, thereby influencing continuous exercise participation.

**Hypothesis 1.** *The degree of autonomy experienced through plogging participation has a positive (+) impact on intrinsic/extrinsic motivation.*

**Hypothesis 2.** *The degree of autonomy experienced through plogging participation has a positive (+) impact on the intention to continue plogging participation.*

**Hypothesis 3.** *Intrinsic/extrinsic motivation mediates the relationship between the experience of autonomy through plogging participation and the intention to continue participation.*

Secondly, competence is a psychological need that is satisfied when an individual's abilities, skills, and talents are appropriately exercised in a specific environment [17]. The desire for competence creates the optimal conditions for demonstrating abilities, and plays a positive role in efforts to maintain and develop skills and abilities through specific activities [17]. In the context of sports, it has been reported that it plays an important role in stimulating intrinsic and extrinsic motivation for continuous training participation for health promotion or sports skill development [39–41]. These research results suggest that a sense of competence positively influences an individual's intrinsic motivation to feel confidence and efficiency, along with extrinsic rewards for acquiring sports skills and exercise abilities. Other previous research has shown that if the desire for competence is not

satisfied due to failure to promote self-regulation or achieve goals in the context of sports participation, it can have a negative impact on intrinsic motivation.

**Hypothesis 4.** *The degree of competence experienced through plogging participation has a positive (+) impact on intrinsic/extrinsic motivation.*

**Hypothesis 5.** *The degree of autonomy experienced through plogging participation has a positive (+) effect on the sustainability of participation.*

**Hypothesis 6.** *Intrinsic/extrinsic motivation mediates the relationship between the experience of autonomy through plogging participation and the sustainability of participation.*

Thirdly, the need for relatedness refers to the desire to feel a sense of belonging and connection with others and to give and receive attention [17]. Relatedness is a fundamental psychological need that is associated with human attachment, which plays a crucial role in maintaining motivation and is particularly important in participating in activities [17,41]. The need for relatedness strengthens intrinsic and extrinsic motivation through stable relationships with meaningful others. From this perspective, numerous studies report that the establishment of stable relationships through interaction with others in exercise or sports activity participation situations has a positive impact on the intention to continue participating in exercise [39–41]. In Korea, plogging activities often take the form of a group of people jogging while picking up trash. Also, SNS posts about plogging activities play a positive role in exposing a positive self-image to others, which is a positive activity in satisfying the need for relatedness.

**Hypothesis 7.** *The degree of relatedness experienced through plogging participation has a positive (+) effect on intrinsic/extrinsic motivation.*

**Hypothesis 8.** *The degree of relatedness experienced through plogging participation has a positive (+) effect on the sustainability of participation.*

**Hypothesis 9.** *Intrinsic/extrinsic motivation mediates the relationship between the experience of relatedness through plogging participation and the sustainability of participation.*

*2.3. An Eco-Friendly Environmental Attitude and Plogging Participation*

As environmental issues have been set as a significant social agenda, the fact that individuals' daily behaviors collectively cause environmental pollution has become widely known, so consumers' socially responsible behaviors and pro-environmental (eco-friendly) consumption habits have begun to take root [42]. Karp [43] defined such socially responsible consumption behavior as self-transcendent behavior that contributes to the welfare of society as a whole rather than individual benefits, and many scholars have conceptualized eco-friendly environmental attitudes and primarily applied them to consumer behavior analysis research [44,45].

Specifically, environmental attitudes refer to an individual's beliefs or values about the environment and the importance of environmental protection [46,47], and many studies related to the execution of eco-friendly behavior commonly suggested that people with high eco-friendly attitudes were more likely to promote decision-making behavior when choosing products or services with environmentally friendly elements compared to people with low eco-friendly environmental attitudes [44,45]. Seo Mun-sik, Eom Sung-won, Son Eun-ji [48] reported that self-determination for pro-environmental consumption was important for consumers of general products to engage in continuous eco-friendly environmental consumption behavior, and that self-determination was regulated according to pro-environmental attitudes.

Plogging participants can be considered a type of consumer who has chosen and participated in plogging, an environmentally friendly sport, among various sports and exercise types. In other words, if plogging is considered a kind of environmentally friendly sports product, then the participants have made various considerations and invested time and effort to execute participatory consumption. In addition, the satisfaction of basic psychological needs that are obtained in this process, together with the stimulation of motivation, affect the intention to participate continuously. In this case, participants with high pro-environmental attitudes are expected to have high self-determination for pro-environmental consumption, and accordingly, the entire decision-making process is strengthened, and the intention to participate continuously is higher. For example, individuals with high pro-environmental attitudes are likely to believe that the degree to which they feel connected to the environment and the community is strong, and that the stable relationship formation with others through plogging participation can be further strengthened through environmental protection activities. In other words, people with high pro-environmental attitudes can be expected to have a stronger intention to continue plogging participation, through the satisfaction of the need for relatedness conveyed by participation experience, than those with low attitudes.

## 3. Research Method

### 3.1. Data Collection

The study aimed to analyze the relationship between the sustained participation intention of plogging participants and the motivational factors mediating self-determination, and to further examine whether this model varies according to the level of formation of eco-friendly attitudes.

For this purpose, we surveyed plogging participants residing nationwide in South Korea using a purposive sampling method, one of the non-probability sampling methods. To collect plogging participants, we selected plogging-related clubs, related organizations, and universities, and conducted online/offline surveys targeting participants who expressed their willingness to participate in the research.

### 3.2. Survey Tools: Measurement Variables

Respondents completed the survey using a self-administration method. To measure the sustained participation intention of participants with plogging activity experience and the factors influencing it, we adapted measurement tools presented in previous studies to suit this study. All independent variables, including three sub-variables of individual psychological needs factors presented in the self-determination theory (autonomy, competence, and relatedness), and eco-friendly attitudes, were measured using Likert's 7-point scale (1: strongly disagree, 7: strongly agree).

To measure sustained participation intention in plogging activities, we adapted the intention to exercise scale presented by Vlachopoulos and Michailidou [49]. We adapted the scales for autonomy, competence, relatedness, intrinsic motivation, and extrinsic motivation developed by Niven and Markland [50] for analyzing self-determination and intrinsic and extrinsic motivation factors influencing continuous walking activity participation. Previous studies such as those by Fenton, Duda, and Barrett and Fletcher [51] also verified and presented that autonomy, competence, and relatedness presented in the self-determination theory had a deep relationship with continuous exercise participation, similar to Niven and Markland [50]. For eco-friendly attitudes, we used the eco-friendly attitude intensity measurement questions presented by Hodgkinson and Innes [52], which have been used in numerous studies measuring eco-friendly consumption intention or eco-friendly activity participation intention. Hodgkinson and Innes [52] restructured the eco-friendly attitude questions after quoting the environmental attitude scale (used in various fields) presented by Dunlap and Van [53] and Forgas and Jolliffe [54], and conducted statistical validity verification. Detailed information about the questions is presented in Table 1.

**Table 1.** Measurement items by variable.

| Item | | Frequency | Source |
|---|---|---|---|
| Sustained participation intention | 1. | I will continue to participate in plogging in the future; | Vlachopoulos, and Michailidou (2006) [49] |
| | 2. | Plogging is important in my life and it will continue to be; | |
| | 3. | I will continue to participate in plogging in any circumstance; | |
| | 4. | In the long term, I believe it is important for me to participate in plogging regularly. | |
| Autonomy | 1. | I believe I can participate in plogging anytime I want; | Niven, Markland, (2016)/ Fenton, Duda and Barrett (2016) [50,51] |
| | 2. | I was able to decide on participating in plogging by myself; | |
| | 3. | I was able to participate in plogging without any significant constraints; | |
| | 4. | I was the one who decided to participate in plogging; | |
| | 5. | I freely chose to participate in plogging of my own volition. | |
| Competence | 1. | I am confident in facing somewhat difficult challenges in plogging; | |
| | 2. | I was confident in my physical ability necessary for performing plogging; | |
| | 3. | I believe I was able to complete somewhat difficult plogging activities; | |
| | 4. | I believe my physical ability has improved through plogging; | |
| | 5. | I feel good thinking that I have the ability to complete plogging. | |
| Relatedness | 1. | I felt an attachment to my colleagues who participate in plogging together; | |
| | 2. | I feel like I am forming a common bond with people who participate in plogging together; | |
| | 3. | I felt camaraderie with people who participate in plogging together; | |
| | 4. | I felt a sense of intimacy with colleagues who have experienced difficulties by plogging together; | |
| | 5. | I felt connected with colleagues during the time of plogging together. | |
| Motivation — Intrinsic | 1. | I think plogging is emotionally fun; | |
| | 2. | I participate in plogging to enjoy the time; | |
| | 3. | I think plogging is a fun activity; | |
| | 4. | I feel joy and satisfaction while plogging. | |
| Motivation — Extrinsic | 1. | I have the motivation to participate in plogging because it brings me financial or other benefits; | |
| | 2. | I was motivated to participate in plogging to receive recognition from friends, family, or colleagues; | |
| | 3. | I have motivation to participate through my plogging, I think I feel the joy of others; | |
| | 4. | I was motivated to participate in plogging due to persuasion from friends, family, or colleagues. | |

**Table 1.** *Cont.*

| Item | Frequency | Source |
|---|---|---|
| Eco-Attitude | 1. I think environmental issues should be prioritized among various social problems;<br>2. Seeing grey clouds over the city makes me feel depressed;<br>3. It annoys me to see people doing nothing for the environment;<br>4. I want to volunteer to help people working to prevent environmental pollution. | Hodgkinson and Innes (2001) [52] |

### 3.3. Data Processing and Analysis

To verify the research hypotheses presented earlier, IBM's AMOS version 28 was utilized to conduct: (1) descriptive statistical analysis, (2) validation of the measurement model, (3) structural equation modeling, and (4) multi-group structural equation analysis.

The appropriateness of the data used in the descriptive statistical analysis was confirmed through the mean and standard deviation, skewness, kurtosis, correlation between independent variables, and item reliability analysis of the measurement variables. Subsequently, in the validation of the measurement model, a confirmatory factor analysis was conducted to determine whether the observed variables adequately explained the latent variables. Specifically, the convergent validity was confirmed through composite reliability (CR), and the discriminant validity was confirmed through the average variance extracted (AVE). This was because the average value of the sub-item measurements was used to measure a single independent variable in this study.

For hypothesis testing, a structural equation model was constructed based on the hypotheses of the preceding theories, and model verification was conducted. Specifically, in this study, a structural equation model including two mediating variables was verified. In this process, the appropriateness of the research model was evaluated, and upon confirmation that the research model was at an appropriate level, the validity, size, and direction of the path coefficients were interpreted. Additionally, a procedure was conducted to decompose the effects of the structural model and analyze the effectiveness of the mediating variables [55].

Finally, the moderating effect of eco-friendly attitudes was verified by analyzing the differences between groups with high eco-friendly attitudes and those without, using multi-group structural equation model analysis. Ultimately, the significance of the difference in path coefficients derived from the two models was verified, and the factors influencing the intention to sustain participation intention were interpreted based on the information obtained.

## 4. Results

### 4.1. Data: Research Subjects

This study surveyed 309 plogging participants residing nationwide in South Korea using a purposive sampling method, one of the non-probability sampling methods. To collect plogging participants, we selected plogging-related clubs, related organizations, and universities, and conducted online/offline surveys targeting participants who expressed their willingness to participate in the research. Out of a total of 309 people, data from 288 were used in the research, excluding 11 people whose response data or content was deemed unsuitable for analysis due to insincerity. All study subjects had participated in plogging activities at least three times, and it was confirmed that the number of plogging participations by the subjects was normally distributed. Detailed information on the research subjects is presented in Table 2.

**Table 2.** General characteristics of the study subjects.

| Item | | Frequency | Percentage (%) |
|---|---|---|---|
| Gender | Male | 165 | 57.3 |
| | Female | 123 | 42.7 |
| Age (years) | ~19 | 60 | 20.8 |
| | 20~29 | 71 | 24.7 |
| | 30~39 | 57 | 19.8 |
| | 40~49 | 41 | 14.2 |
| | 50~59 | 40 | 13.9 |
| | 60~ | 19 | 6.6 |
| Plogging experience (years) | ~5 | 105 | 36.5 |
| | 5~7 | 70 | 24.3 |
| | 7~ | 113 | 39.2 |
| Participation type | Regular | 230 | 78.4 |
| | Irregular | 58 | 21.6 |
| Region | Metropolitan | 142 | 49.3 |
| | Non-metropolitan | 146 | 50.7 |
| Income | ~USD 1800 | 156 | 54.2 |
| (Monthly) | USD 1800~3700 | 62 | 21.5 |

*4.2. Descriptive Statistics, Correlation Analysis by Factor, and Reliability Analysis*

Table 3 presents the descriptive statistics of the main variables measured in this study. All measured variables are continuous, and their mean, standard deviation, minimum value, maximum value, skewness, and kurtosis were examined. The mean values of the constructs indicate that the intention to sustain plogging was 4.56 (sd = 1.83), autonomy was 4.98 (sd = 1.75), competence was 4.92 (sd = 1.54), relatedness was 4.88 (sd = 1.64), intrinsic motivation was 4.62 (sd = 1.60), extrinsic motivation was 4.50 (sd = 1.56), and eco-friendly attitude was 5.11 (sd = 2.04). To verify the normality of the main variables, the skewness and kurtosis values were checked. Skewness showed values from −0.48 to −0.068, and kurtosis showed values from −0.48 to 0.12, confirming that they satisfy normality (West, Finch and Curran, 1995 [55]).

**Table 3.** Descriptive statistics of variables.

| Variable | M | SD | Min | Max | Skewness | Kurtosis |
|---|---|---|---|---|---|---|
| SPI | 4.56 | 1.83 | 1 | 7 | −0.313 | −0.228 |
| AUT | 4.98 | 1.75 | 1 | 7 | −0.450 | 0.125 |
| COM | 4.92 | 1.54 | 1 | 7 | −0.410 | −0.298 |
| REL | 4.88 | 1.64 | 1 | 7 | −0.482 | 0.250 |
| MI | 4.62 | 1.60 | 1 | 7 | −0.068 | −0.097 |
| EI | 4.50 | 1.56 | 1 | 7 | −0.322 | −0.358 |
| EA | 5.11 | 2.04 | 1 | 7 | −0.377 | −0.489 |

Table 4 presents the results of the correlation analysis for the main variables of this study. If the correlation coefficient between the constructs used in multiple regression analysis exceeds 0.7, multi-collinearity can occur, leading to statistical errors. However, the correlation coefficients among the main variables of this study were all at a level lower than 0.7, confirming that there were no issues with statistical errors due to multi-collinearity.

**Table 4.** Correlation analysis between independent variables.

| | 1 | 2 | 3 | 4 | 5 | 6 | 7 |
|---|---|---|---|---|---|---|---|
| 1. SPI | - | | | | | | |
| 2. AUT | 0.673 | | | | | | |
| 3. COM | 0.679 | 0.652 | | | | | |
| 4. REL | 0.691 | 0.623 | 0.692 | | | | |
| 5. MI | 0.697 | 0.622 | 0.639 | 0.683 | | | |
| 6. EI | 0.419 | 0.336 | 0.324 | 0.584 | 0.442 | | |
| 7. EA | 0.504 | 416 | 0.392 | 0.489 | 0.462 | 0.378 | |

### 4.3. Assessment of Measurement Model: Reliability and Validity Analysis

In this study, Cronbach's α test and confirmatory factor analysis (CFA) were conducted to verify the reliability and validity of the measurement tools used. Particularly, as the reliability and validity of the measurement tool can vary depending on the subject, this study not only considered the entire group but also divided the group into those with high and low eco-friendly attitudes to verify whether the measured tools for each group were at a satisfactory level. For eco-friendly attitudes, the average value was 5.11, and the group was divided (mean split) based on this average.

Confirmatory factor analysis was conducted to analyze the fit of the measurement model. The results showed that the entire group ($\chi 2$ = 198.822 [df = 105, $p$ < 0.01], $\chi 2/\mathrm{df}$ = 1.893, TLI = 0.927, CFI = 0.944, RMSEA = 0.058, SRMR = 0.062), the group with high eco-friendly attitudes ($\chi 2$ = 211.454 [df = 105 < 0.01], $\chi 2/\mathrm{df}$ = 2.013, TLI = 0.897, CFI = 0.881, RMSEA = 0.062, SRMR = 0.069), and the group with low eco-friendly attitudes ($\chi 2$ = 231.778 [df = 105 < 0.01], $\chi 2/\mathrm{df}$ = 2.207, TLI = 0.847, CFI = 0.834, RMSEA = 0.064, SRMR = 0.067) all exhibited satisfactory levels.

Upon conducting the reliability analysis in Table 5, the reliability α values of each sub-domain for the entire group were between 0.856 and 0.917, for the group with high eco-friendly attitudes they were between 0.856 and 0.911, and for the group with low eco-friendly attitudes, they were between 0.877 and 0.914. Since a reliability level of 0.6 or above is considered satisfactory, it was confirmed that the internal reliability of the items used in this study were all at a trustworthy level.

**Table 5.** Reliability analysis of measurement items (CFA).

| Variable | b | SE | t | α | | | CR | | | AVE | | |
|---|---|---|---|---|---|---|---|---|---|---|---|---|
| | | | | Total | EA(H) | EA(L) | Total | EA(H) | EA(L) | Total | EA(H) | EA(L) |
| AUT 1 | 1.000 | - | - | 0.917 | 0.911 | 0.924 | 0.950 | 0.898 | 0.778 | 0.863 | 0.747 | 0.540 |
| AUT 2 | 0.981 | 0.031 | 21.06 | | | | | | | | | |
| AUT 4 | 0.905 | 0.033 | 19.55 | | | | | | | | | |
| COM 2 | 1.000 | - | - | 0.909 | 0.900 | 0.886 | 0.875 | 0.793 | 0.812 | 0.701 | 0.561 | 0.591 |
| COM 3 | 0.895 | 0.066 | 30.01 | | | | | | | | | |
| COM 5 | 0.756 | 0.051 | 17.91 | | | | | | | | | |
| REL 1 | 1.000 | - | - | 0.903 | 0.873 | 0.910 | 0.835 | 0.831 | 0.807 | 0.629 | 0.623 | 0.584 |
| REL 2 | 0.815 | 0.037 | 18.66 | | | | | | | | | |
| REL 3 | 0.701 | 0.020 | 18.89 | | | | | | | | | |
| MI 1 | 1.000 | - | - | 0.911 | 0.856 | 0.896 | 0.866 | 0.845 | 0.799 | 0.683 | 0.647 | 0.572 |
| MI 2 | 0.853 | 0.030 | 20.35 | | | | | | | | | |
| MI 3 | 0.768 | 0.067 | 17.53 | | | | | | | | | |
| EI 1 | 1.000 | - | - | 0.856 | 0.889 | 0.877 | 0.752 | 0.812 | 0.784 | 0.504 | 0.591 | 0.548 |
| EI 2 | 0.749 | 0.056 | 15.64 | | | | | | | | | |
| EI 3 | 0.657 | 0.075 | 16.35 | | | | | | | | | |

Note 1. Total model fit: $\chi 2$ = 198.822 (df = 105, $p$ < 0.01), $\chi 2/\mathrm{df}$ = 1.893, TLI = 0.927, CFI = 0.944, RMSEA = 0.058, SRMR = 0.062. Note 2. EA high group: $\chi 2$ = 211.454(df = 105 < 0.01), $\chi 2/\mathrm{df}$ = 2.013, TLI = 0.897, CFI = 0.881, RMSEA = 0.062, SRMR = 0.069. Note 3. EA low group: $\chi 2$ = 231.778 (df = 105 < 0.01), $\chi 2/\mathrm{df}$ = 2.207, TLI = 0.847, CFI = 0.834, RMSEA = 0.064, SRMR = 0.067.

The CR value for verifying the convergent validity of the measurement tool was 0.752–0.950 for the entire group, 0.793–0.898 for the group with high eco-friendly attitudes, and 0.778–0.812 for the group with low eco-friendly attitudes. Considering that the commonly accepted figure is 0.5, it was confirmed that the measurement tool of this study was at an appropriate level [56]. The AVE value, which was confirmed to verify the discriminant validity of the measurement tool, was 0.504–0.863 for the entire group, 0.561–747 for the group with high eco-friendly attitudes, and 0.540–591 for the group with low eco-friendly attitudes. Considering that the commonly accepted figure is between 0.5 and 0.95, it was confirmed that the measurement tool of this study would secure discriminant validity [57].

### 4.4. The Role of Motivation in Mediating the Relationship between Self-Determination Factors and Sustained Participation Intention: Structural Equation Analysis (Study 1)

This study conducted a structural equation modeling analysis to examine the relationship between the degree of self-determination factors experienced by plogging participants, their intrinsic and extrinsic motivation, and their intention to continue participating. Based on the hypotheses of research question 1 (Hypotheses 1–9), we constructed and analyzed the research model. First, the fit of the research model was analyzed and the results showed that $X2(df) = 479.08(219)$, TLI = 0.915, CFI = 0.903, and RMSEA = 0.056. This indicated that the structural equation model was at an acceptable level for interpretation [58].

Looking at the path process of the structural model presented in [Figure 1] in detail, the basic psychological needs factors of autonomy (b = 0.422, $p < 0.001$), competence (b = 0.214, $p < 0.05$), and relatedness (b = 0.681, $p < 0.001$), proposed in the self-determination theory, were found to have a positive (+) impact on intrinsic motivation. This suggests that the stronger the experience of autonomy and competence in past plogging participation and the better the relatedness with the people who participated together, the stronger the intrinsic motivation for plogging participation. Among these, the experience of relatedness had the greatest impact on intrinsic motivation, and the experience of competence was analyzed as the lowest. Conversely, extrinsic motivation showed a somewhat different trend, where the intensity of the experience of autonomy (b = 0.155, $p > 0.05$) and competence (b = 0.021, $p > 0.05$) did not have a significant impact, and the experience of relatedness with the people who participated together (b = 0.245, $p < 0.01$) had a significantly positive (+) impact. This suggests that the relationship between the formation of plogging exercise participation motivation, according to the type and degree of basic human needs satisfaction proposed in the self-determination theory, is mainly related to intrinsic motivation, and for extrinsic motivation, only relatedness has an impact.

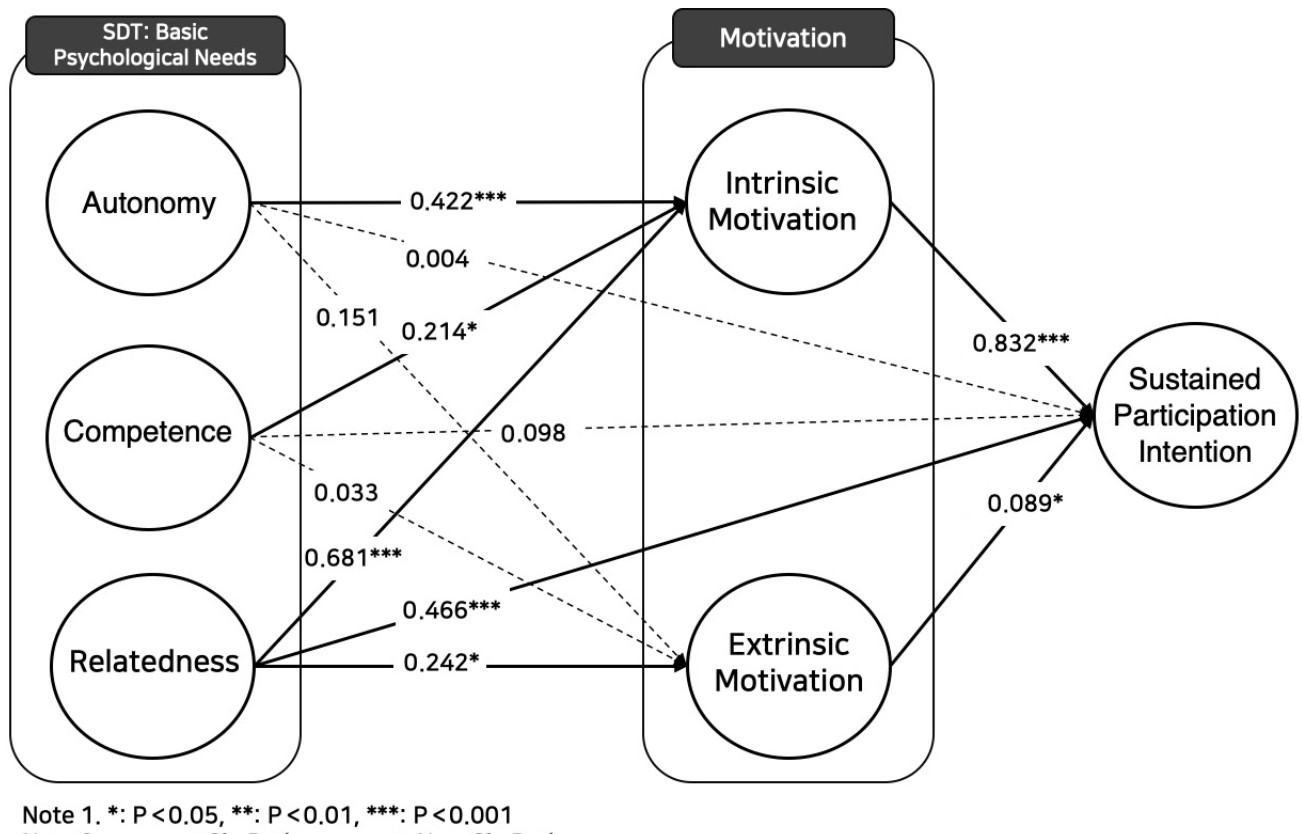

**Figure 1.** Verification of the structural equation model.

In contrast, when analyzing the direct effect of the basic psychological needs satisfaction factors proposed in the self-determination theory on the relationship with the intention to continue plogging participation, the experience of autonomy (b = 0.004, *p* > 0.05) and competence (b = 0.098, *p* > 0.05) did not have a significant impact, whereas the experience of relatedness with the people who participated together (b = 0.466, *p* < 0.001) had a direct positive (+) effect. In addition, the intrinsic motivation (b = 0.832, *p* < 0.001) and extrinsic motivation (b = 0.089, *p* < 0.05) set as mediators both had a statistically significant positive (+) impact on the intention to continue participating, with intrinsic motivation having the greater impact.

In addition, an effect decomposition analysis was conducted to analyze the effect of intrinsic and extrinsic motivation mediating the relationship between the basic psychological needs satisfaction experience through plogging activities and the intention to continue plogging participation. The analysis results are presented in Table 6.

**Table 6.** Decomposition of effects in the structural model.

| Path | Direct Effect | Indirect Effect | Total Effect | Bias Corrected, 95% (BC) |
|---|---|---|---|---|
| REL → MI → SPI | 0.466 | 0.566 | 1.032 | 0.178~0.397 *** |
| REL → ME → SPI | 0.466 | 0.021 | 0.487 | 0.012~0.201 * |

Note 1. The mediating effect of intrinsic/extrinsic motivation between autonomy and intention to continue participating does not hold. Note 2. The mediating effect of intrinsic/extrinsic motivation between competence and intention to continue participating does not hold. Note2. *: *p* < 0.05, ***: *p* < 0.001.

In the case of autonomy, the direct effect on the intention to continue plogging participation and extrinsic motivation was not significant, and only a direct effect on intrinsic motivation was observed. Furthermore, since intrinsic motivation had a direct impact on the intention to continue participating, the mediation model of intrinsic and extrinsic motivation mediating the impact of autonomy on the intention to continue participating did not hold because the direct effect of autonomy on the intention to continue participating was not significant (b = 0.004, *p* > 0.05). Similarly, in the case of competence, the direct effect on the intention to continue participating was not significant, so the mediation model did not hold (b = 0.098, *p* > 0.05). In contrast, in the case of relatedness, the direct effect on the intention to continue participating was significant (b = 0.466, *p* < 0.001), and the effects of relatedness on both intrinsic motivation (b = 0.681, *p* < 0.001) and extrinsic motivation (b = 0.242, *p* < 0.05) were significant. Since both intrinsic motivation (b = 0.832, *p* < 0.001) and extrinsic motivation (b = 0.089, *p* < 0.05) had significant effects on the intention to continue participating, the mediation model held. Specifically, the indirect effect of intrinsic motivation mediating the relationship between relatedness and the intention to continue participating was 0.566 (B.C: 0.178~0.397, *p* < 0.001), and the indirect effect of extrinsic motivation was 0.021 (B.C: 0.012~0.201, *p* < 0.001). The indirect effect of intrinsic motivation was greater than the direct effect of relatedness on the intention to continue participating, and the indirect effect of extrinsic motivation was smaller. The sum of the indirect effects of intrinsic and extrinsic motivation (0.587) was greater than the direct effect of relatedness on the intention to continue participating (0.466), indicating that intrinsic and extrinsic motivation fully mediate the relationship between relatedness and the intention to continue participating. However, the fact that the indirect effect of extrinsic motivation (0.021) was significantly smaller than that of intrinsic motivation (0.566) confirms the important role of intrinsic motivation.

*4.5. Verification of Group Differences According to Eco-Friendly Attitudes: Multi-Group Structural Equation Analysis (Study 2)*

This study performed a multi-group structural equation analysis, to verify the differences in motivation, which mediates the relationship between the degree of perceived basic psychological needs satisfaction and the intention to continue participation through plogging participation analyzed earlier, according to eco-friendly attitudes. The average

value of eco-friendly attitudes was 5.11, and the group with a lower average was named the low eco-friendly attitude group (146 people), and the group with a higher average was named the high eco-friendly attitude group (142 people).

To compare the path coefficients between the two groups, we analyzed the similarity between the two models by comparing the form similarity and structure similarity. The results showed that the unconstrained model and the form similarity model exhibited a good fit with $\chi2 = 524.375$ (df = 187), CFI = 0.915, and RMSEA = 0.051. In the measurement similarity model, when the path between the latent variable and the measurement variable was constrained to be the same, $\chi2 = 589.758$ (df = 200), CFI = 0.913, and RMSEA = 0.048, the model and data showed a relatively good fit and there was no statistically significant difference from the form similarity model. Through this, it was confirmed that the observed variables measuring each construct were perceived equal between groups. In addition, in the structural similarity model, when the variance and covariance of the latent variables were constrained, it was analyzed as $\chi2 = 605.217$ (df = 218), CFI = 0.910, and RMSEA = 0.044, and the model and data showed a relatively good fit. Furthermore, there was a statistically significant difference in the $\chi2$ difference comparison between the form similarity and measurement similarity models. Based on these results, it was confirmed that the two groups, divided according to eco-friendly attitudes, had an effect as control variables, and a path coefficient comparison analysis was performed for the high and low groups of eco-friendly attitudes. The analysis results are shown in Table 7.

**Table 7.** Estimated path coefficients and comparison by group.

| Path | Δχ2 | EA (Low) | | EA (High) | |
|---|---|---|---|---|---|
| | | b | SE | b | SE |
| AUT → MI | 0.94 | 0.402 *** | 0.102 | 0.500 *** | 0.088 |
| AUT → ME | 3.09 ** | 0.243 * | 0.143 | 0.113 | 0.160 |
| AUT → SPI | 0.55 | 0.073 | 0.097 | 0.085 | 0.084 |
| COM → MI | 4.01 ** | 0.194 ** | 0.118 | 0.054 | 0.114 |
| COM → ME | 3.09 ** | 0.001 | 0.167 | 0.056 | 0.211 |
| COM → SPI | 0.08 | 0.027 | 0.099 | 0.140 | 0.087 |
| REL → MI | 0.98 | 0.586 *** | 0.125 | 0.633 *** | 0.110 |
| REL → ME | 2.79 * | 0.172 * | 0.174 | 0.328 * | 0.199 |
| REL → SPI | 0.43 | 0.563 *** | 0.124 | 0.412 *** | 0.102 |
| MI → SPI | 0.98 | 0.386 *** | 0.100 | 0.447 *** | 0.089 |
| EI → SPI | 0.99 | 0.104 * | 0.063 | 0.028 | 0.045 |

Note 1. Indirect effect between AUT → MI → SPI (BC 95%): 0.047~0.191 **. Note 2. Indirect effect between COM → MI → SPI (BC 95%): 0.051~0.301 **. Note 3. Indirect effect between REL → MI/ME → SPI (BC 95%): 0.009~0.071 ***. Note 3. *: $p < 0.05$, **: $p < 0.01$ ***: $p < 0.001$.

First, in the case of the influence of autonomy on intrinsic motivation, both the low eco-friendly attitude group (b = 0.402, $p < 0.001$) and the high group (b = 0.500, $p < 0.001$) had a significant positive influence, and no difference was found in the two path coefficients ($\Delta\chi2 = 0.94$, $p > 0.05$). In the case of the influence of autonomy on extrinsic motivation, while the low eco-friendly attitude group had a significant influence (b = 0.243, $p < 0.05$), the high eco-friendly attitude group did not have a statistically significant influence (b = 0.113, $p > 0.05$). The analysis of the difference in path coefficients was statistically significant ($\Delta\chi2 = 3.09$, $p < 0.01$). In the case of the influence of autonomy on the intention to continue participation, neither the high eco-friendly attitude group (b = 0.073, $p > 0.05$) nor the low group (b = 0.085, $p > 0.05$) had a significant influence, and the difference in the two path coefficients was also not significant ($\Delta\chi2 = 0.55$, $p > 0.05$). In the case of the influence of competence on intrinsic motivation, while the low eco-friendly attitude group (b = 0.194, $p < 0.01$) was significant, the high eco-friendly attitude group (b = 0.054, $p > 0.05$) was not significant, and the difference in the two path coefficients was also significant ($\Delta\chi2 = 0.4.01$). In the case of the influence of competence on extrinsic motivation, neither the low eco-friendly attitude group (b = 0.001, $p > 0.05$) nor the high group (b = 0.056, $p > 0.05$) had a significant influence, and no difference

was found between the two path coefficients ($\Delta\chi2 = 0.080$, $p > 0.05$). In the case of relatedness, both the low eco-friendly attitude group (b = 0.586, $p < 0.001$) and the high eco-friendly group (b = 0.633, $p < 0.001$) had a significant influence on intrinsic motivation. Both the low eco-friendly attitude group (b = 0.563, $p < 0.001$) and the high eco-friendly attitude group (b = 0.412, $p < 0.001$) had a significant influence on the intention to continue participation. However, only the high eco-friendly attitude group (b = 0.328, $p < 0.05$) had a significant influence on extrinsic motivation. The path coefficient difference verification results showed no significant differences between the two groups for the influence on intrinsic motivation ($\Delta\chi2 = 0.98$, $p > 0.05$) and the influence on the intention to continue participation ($\Delta\chi2 = 0.43$, $p > 0.05$), but a significant difference was confirmed between the two groups in the case of extrinsic motivation ($\Delta\chi2 = 2.79$, $p < 0.01$).

To understand the different patterns of mediating effects that can occur depending on eco-friendly attitudes, we compared the mediating effect analysis for each model of the low and high eco-friendly attitude groups (Table 8). First, in the case of the low eco-friendly attitude group, the direct effect of autonomy on the intention to continue participation was not significant (b = 0.55, $p > 0.05$), so the mediating effects of intrinsic motivation and extrinsic motivation mediating autonomy and the intention to continue participation were not established. The same was true for competence, where the direct effect on the intention to continue participation (b = 0.027, $p > 0.05$) was not significant, so the mediating effects of intrinsic motivation and extrinsic motivation were not established. It was confirmed that there were significant correlations between relatedness and intrinsic motivation (b = 0.633, $p < 0.001$), relatedness and extrinsic motivation (b = 0.172, $p < 0.05$), intrinsic motivation and the intention to continue participation (b = 0.447, $p < 0.001$), and extrinsic motivation and the intention to continue participation (b = 0.104, $p < 0.05$). Also, the relationship between relatedness and the intention to continue participation (b = 0.412, $p < 0.001$) was significant. Through this, it was confirmed that the model in which intrinsic and extrinsic motivation mediated relatedness and the intention to continue participation was established, and the significance of the mediating effect was also confirmed through the Sobel test (B.C: 0.012~0.131, $p < 0.001$). Specifically, it was confirmed that only intrinsic motivation partially mediated the relationship between relatedness and the intention to continue participation, considering that the influence of relatedness on extrinsic motivation was not significant and that the direct effect (b = 0.563) of relatedness on the intention to continue participation was greater than the indirect effect (b = 0.244) that influenced through intrinsic motivation.

**Table 8.** Decomposition of effects in the structural model by generational groups.

| Group | Path | Direct Effect | Indirect Effect | Total Effect | Bias Corrected, 95% (BC) |
|---|---|---|---|---|---|
| EA (Low) | REL → SPI | 0.563 | 0.244 | 0.789 | 0.012~0.131 *** |
| EA (High) | REL → SPI | 0.412 | 0.292 | 0.704 | 0.049~0.205 ** |

\*\*: $p < 0.01$, \*\*\*: $p < 0.001$.

In both the high and low eco-friendly attitude groups, the mediating effects of intrinsic and extrinsic motivation were not established because there was no significant correlation between the intention to continue participation and autonomy and competence. However, a somewhat different pattern of mediating effects was found in relatedness. It was confirmed that there were significant correlations between relatedness and intrinsic motivation (b = 0.633, $p < 0.001$), and between intrinsic motivation and the intention to continue participation (b = 0.447, $p < 0.001$). In addition, the relationship between relatedness and the intention to continue participation (b = 0.412, $p < 0.001$) was significant. However, the relationship between extrinsic motivation and the intention to continue participation (b = 0.028, $p > 0.05$) was not significant. Through this, it was confirmed that the model in which intrinsic motivation mediated relatedness and the intention to continue participation was established, and the significance of the mediating effect was also confirmed through the Sobel test (B.C: 0.049~0.205,

*p* < 0.001). Specifically, it was confirmed that only intrinsic motivation partially mediated the relationship between relatedness and intention.

## 5. Discussion and Conclusions

Eco-friendly sports emphasize the interaction between humans and nature, allowing us to reconsider our lifestyles and responsibilities towards the environment. They also play a positive role in individual health and social life as healthy exercise [59]. This study analyzed the relationship between the three basic psychological needs of a human: autonomy, competence, and relatedness, intrinsic and extrinsic motivation, and the intention to continue participating in plogging, an eco-friendly sport. The study also analyzed the moderating effect of eco-friendly attitudes on the previously established model to understand the uniqueness of plogging activities as eco-friendly behaviors. Based on the theoretical background of self-determination theory and previous studies related to eco-friendly attitudes, a total of 10 hypotheses were set and verified.

First, Hypotheses 1–3, which analyzed the relationship between the fulfillment of autonomy and the intention to continue participating and intrinsic/extrinsic motivation, were partially supported for Hypothesis 1, but Hypotheses 2 and 3 were rejected. In the case of Hypothesis 1, the promotion of intrinsic motivation for plogging according to the degree of autonomy experienced through plogging participation was found, but extrinsic motivation was not found, so it was partially supported. In the case of Hypothesis 2, the influence of autonomy experienced through the intention to continue plogging was not significant, so it was rejected. Therefore, the mediating effect of intrinsic/extrinsic motivation mediating the relationship between autonomy and the intention to continue participating was not established, so Hypothesis 3 was also rejected. Although this supports the results of previous studies [60,61], in that the fulfillment of autonomy affects intrinsic motivation, the results differed in that it does not affect extrinsic motivation or have a direct impact on the intention to continue participating. This may be because plogging is related to volunteer activities as an eco-friendly sport, not a simple concept of healthy exercise. Volunteer activities are altruistic behaviors that are unrelated to extrinsic rewards. Therefore, it can be interpreted that the impact on extrinsic motivation was minimal. The fact that no difference in participation intention was found according to the degree of autonomy experience may be due to the small variance; specifically, most plogging participants did not feel a compulsion in the participation experience as volunteers, and, accordingly, no difference in the intention to continue participating was found.

Next, Hypotheses 4–6, which verified the relationship between competence and the intention to continue participating and intrinsic/extrinsic motivation, were partially supported for 4 but rejected for 5 and 6. Generally, the experience of competence is the factor that most influences the intention to continue participating in healthy exercise or sports activities [39–41]. Exercise participants feel a sense of competence in the process and experience of achieving goals such as skeletal muscle enhancement and weight loss through exercise participation [62]. Furthermore, sports activity participants feel a sense of competence through experiences of victory in competition or record reduction. This sense of competence has a strong influence on intrinsic motivation, such as pleasure and satisfaction, and also affects extrinsic motivation by acting as a reward. However, in the results of this study, competence did not directly influence the intention to continue exercising and extrinsic motivation, but only influenced intrinsic motivation, partially supporting the results of previous studies. A possible reason is that the participation motivation and experience of eco-friendly sports participants are different from general exercise for health activities. The results of this study coincide with numerous studies explaining eco-friendly consumer behavior, which argue that the intrinsic reward system does not operate in altruistic behavior, and that self-efficacy and the satisfaction obtained through environmental protection activities act as important motivations [44,45,48].

Third, Hypotheses 7–9, which verified the relationship between relatedness and intrinsic/extrinsic motivation and the intention to continue participating, were all supported, unlike

the previous results. As discussed earlier, relatedness is a basic psychological need defined by the desire to feel a sense of belonging by connecting with others and society and plays an important role in maintaining motivation [41]. While autonomy and competence only influenced intrinsic motivation and did not influence the intention to continue participating, relatedness influenced both intrinsic and extrinsic motivation, as well as the intention to continue participating. This result is consistent with the results of previous studies which showed that group participation was important in plogging, an eco-friendly sport, and played an important role in positive self-image exposure using social networks [7,9]. In this study, the partially mediating effects of intrinsic and extrinsic motivation were confirmed, but from the decomposition of the total effect, the effect of extrinsic motivation converged close to 0, confirming the importance of the role of intrinsic motivation.

Lastly, Hypothesis 10, which verified the moderating effect of eco-friendly attitudes, was also supported because some path differences in the structural model were found between the group with high eco-friendly attitudes and the group with low eco-friendly attitudes. In particular, there were differences in the path coefficients for the influence of autonomy on extrinsic motivation, the influence of competence on intrinsic and extrinsic motivation, and the influence of relatedness on extrinsic motivation. In the group with high eco-friendly attitudes, there was an effect of increasing extrinsic motivation according to the experience of autonomy, but no significant correlation was found in the group with low eco-friendly attitudes. In addition, competence did not influence either intrinsic or extrinsic motivation in the group with high eco-friendly attitudes. In other words, people with high eco-friendly attitudes understand that the intention to continue participating in plogging increases according to intrinsic motivation, regardless of the experience of autonomy or competence. The study results imply that the unique exercise effect of plogging, as suggested by Raghavan, Panicker and Emmatty [11], does not influence continuous participation, and the strategy to increase the intention to continue participating through the promotion of exercise effects may not be effective.

In summary, the study results support those of previous studies that have researched the relationship between the fulfillment of basic human needs such as autonomy, competence, and relatedness, as suggested by self-determination theory, and the intention to participate in sports activities, but also derive some different results. In particular, the results are different from previous studies in that while the experiences of autonomy, competence, and relatedness influence intrinsic motivation and affect the intention to continue participating, autonomy and competence do not affect extrinsic motivation, and relatedness operates as the most influential explanatory variable. These results confirm that plogging activities have a strong concept of volunteer activities, and the role of plogging activities is more related to pro-social behavior for enhancing relationships with others than to the physical competence gained from participation. It also confirms that autonomy plays an important role through it being a volunteer activity. Through this, it may be tentatively concluded that voluntary participation and socializing activities are the core values conveyed by plogging activities. In addition, the study results confirm that plogging is a volunteer activity that provides a sporting experience and an eco-friendly behavior, and the value felt by the participants and the effect of the various influencing factors on the intention to continue participating are different depending on the formation level of their eco-friendly attitudes.

The results of this study have the following academic and policy implications. First, from an academic perspective, this study contributes to understanding the role of self-determination factors for continuing participation in physical activities such as plogging from a theoretical perspective. Reconfirming the importance of the experience of autonomy, competence, and relatedness obtained through plogging participation and understanding how these experiences influence the intention to continue plogging through intrinsic/extrinsic motivation, as well as attempting an interdisciplinary approach to promote environmentally friendly behavior in the sports sector, also has academic significance in that it contributes to reducing the theoretical gap in the relationship between eco-friendly sports participation behavior, attitudes, and motivation. Many previous studies point out the situation that even if there

is consideration for or valuing of the environment, it is not often directly connected to eco-friendly behavior or eco-friendly consumption, and mention the need for research to reduce the gap between consideration for or valuing of the environment and eco-friendly behavior [63,64]. In this regard, this study contributes to reducing the theoretical gap in the relationship between eco-friendly sports participation behavior, attitudes, and motivation, which is also of academic significance.

Furthermore, from a policy perspective, the study results have significance in that they can be used as the basis for developing policies to develop and support eco-friendly sports to promote physical activity and environmental protection. From a sociological perspective, participation in eco-friendly sports contributes to raising the environmental awareness of society members, and these activities promote community participation and social responsibility for the environment [65]. Therefore, the study results provide the information necessary for policymakers to encourage and maintain plogging participation, which is meaningful. That is, through informed policy-making, plogging can play a role in creating a healthier and more sustainable society and environment while improving the physical and mental health of society members.

**Author Contributions:** Writing—original draft, J.K.; Writing—review & editing, S.K.; Supervision, J.C.; Project administration, J.K. All authors have read and agreed to the published version of the manuscript.

**Funding:** This research received no external funding.

**Institutional Review Board Statement:** The study was conducted in accordance with the Declaration of Helsinki, and approved by the Institutional Review Board of Bioethics Review Committee of Dongguk University.

**Informed Consent Statement:** Informed consent was obtained from all subjects involved in the study.

**Data Availability Statement:** This study does not contain personally identifiable information. However, data availability may be limited due to the absence of individual consent for public sharing.

**Conflicts of Interest:** The authors declare no conflict of interest.

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
