# Peer review of "Examining the Relationship between Pro-Environmental Attitudes, Self-Determination, and Sustained Intention in Eco-Friendly Sports Participation: A Study on Plogging Participants"

_sustainability, doi:10.3390/su151511806_

Round 1

Reviewer 1 Report

After reviewing the article Examining the Relationship between Pro-Environmental Attitudes, Self-Determination, and Sustained Intention in Eco-Friendly Sports Participation: A Study on Plogging Participants, I consider it suitable for publication for several reasons that are inherent to it. Firstly, it should be noted that the article has a consolidated theoretical foundation that serves to support and understand the objective and hypotheses put forward by the authors. In this sense, it should be noted that the Survey Tools: Measurement Variables proposed are adequate to find results that are binding to the article, such as the fact that this study will contribute to understanding the role of the factors, self-determination factors that continue participation in physical activities such as plogging from a theoretical perspective, among others. Also, in the conclusion and discussion section, all the hypotheses put forward by the authors are validated, and the main findings are clearly and concisely presented. Likewise, the bibliographical references are well cited, according to the standards proposed by the journal. 

Author Response

Thank you for taking the time to review our manuscript, "Examining the Relationship between Pro-Environmental Attitudes, Self-Determination, and Sustained Intention in Eco-Friendly Sports Participation: A Study on Plogging Participants." We appreciate your constructive feedback and are gratified that you consider our work suitable for publication.

We are particularly pleased to note your acknowledgement of our consolidated theoretical foundation. We endeavored to build a robust theoretical framework that effectively supports and explains our research objective and hypotheses. Your recognition of the appropriateness of our survey tools and measurement variables is also greatly valued, as they were carefully selected to yield meaningful results concerning self-determination factors influencing sustained participation in eco-friendly sports like plogging.

Furthermore, we are heartened that you found our conclusion and discussion section comprehensive and clear. We aimed to validate our hypotheses and present our main findings in a succinct and coherent manner, and it is reassuring to learn that we have been successful in these respects.

Lastly, we are grateful for your observation that our bibliographic references were well cited according to the journal's standards. We are committed to adhering to these standards and maintaining the academic integrity of our work.

Once again, thank you for your careful evaluation and favorable endorsement of our manuscript. We look forward to making any further revisions if required and hope for the successful publication of our work in your esteemed journal.

Best regards,

Reviewer 2 Report

Please find the following information meant to assist in the refining and crafting of the manuscript. I recognize and applaud the hard work the was put into the conducting of this research and production of this manuscript. The information below comes from a place of interest in helping, not one of judgement or condescension. 

Overall, I think this is a product that can be refined into a fine piece of scientific literature. In its present form, I do not believe it is appropriate for publication. That said, this is one heck of a draft that can be modified to elevate the overall quality. 

Abstract – (major adjustments need to be made in this section. I do not believe they are difficult, but at paramount to the manuscript). 

Should this not be written in past tense?

Please consider placing more succinct information in the abstract. How many participants were surveyed? Add numerical results. 

Results indicate, the study cannot find. Authors conclude, the study cannot conclude. 

Please check to confirm the word count is within the manuscript parameters. 

Introduction – 

In this context, many members of society have become more aware of the impact of 31 their actions on the planet, leading to the rise of terms such as "environment", "eco-32 friendly", "sustainability", and "green" as significant societal agendas [2].

= The intent of this sentence is understood, but I do not believe the grammar is correct. 

Seek consistent in text citation format. 

Backdrop? A lovely word, but I question is applicability here. 

Line 39 and 42– avoid ‘This’. Be specific in your use of words. If a future author is to draw this sentence from this piece, the term ‘This’ can be construed. 

Past tense – please frame the text in past tense. 

Theoretical Background and Research Hypotheses

Please be sure to have every pointed statement (a statement that expresses factual information) has a citation and accompanying reference. 

Why is eco-friendly bolded? 

Research Method

Was IRB approval earned? 

Should the first table not be in the results section? 

Results

Remove the brackets at the beginning of the paragraphs – [Table X].

Line 477 – If you have a First, you must have a Second. 

I really like the SEM section. 

5. Discussion and Conclusion

My overall recommendation for this section is to tighten up the language being used. 

As difficult and ‘unromantic’ as it may be, being precise and concise with wording is the appropriate tact to take. State the facts. Reduce any lead-in terms (First, Additionally, Similarly). Personally, I find this the most difficult aspect of writing manuscripts. I want to explain the research and convey what I believe to be exciting or interesting. But….this type of document is meant to offer information – not tell a story as such. Think Cubist versus Impressionist – both portray a subject, but one does it in succinct bits and another with graceful nuance. Be a Cubist. 

Avoid vague terms (This, They, The study). This = can be specified. They = the participants, specific researchers. The study – which study in particular?. 

Author Response

Dear Reviewer 2,

Thank you very much for taking the time to provide an in-depth review of our manuscript, "Examining the Relationship between Pro-Environmental Attitudes, Self-Determination, and Sustained Intention in Eco-Friendly Sports Participation: A Study on Plogging Participants.“

We genuinely appreciate your constructive criticism and find your insights invaluable in refining our work. We wholeheartedly agree that feedback, like yours, contributes significantly to the improvement of our manuscript rather than being a form of judgment or condescension.

We are encouraged by your acknowledgement of our efforts in conducting this research, and your belief that it has the potential to become a fine piece of scientific literature, despite not being ready for publication in its current form. Your perspective gives us confidence that with further refinement, our manuscript can reach the quality required for publication.

Please rest assured that we will take all your suggestions and comments into serious consideration during our revision process. Your detailed and insightful feedback will undoubtedly assist us in enhancing the overall quality of our manuscript.

From this point forward, we will be detailing the changes and corrections we have made in response to your comments. Please be assured that we have endeavored to do our utmost in making these revisions.

Once again, we extend our sincere thanks for your time and effort in reviewing our work. Your continued guidance and feedback are invaluable as we navigate through the revision process.

With sincere regards,

1. Abstract

1) Should this not be written in past tense?

>> We have revised the abstract to be in the past tense to accurately reflect that the research and analysis have already been conducted.

2) Please consider placing more succinct information in the abstract. How many participants were surveyed? Add numerical results.

>> We have added succinct information about the number of participants surveyed in our study. We now specify that a total of 288 participants were randomly assigned to the survey.

3)  Results indicate, the study cannot find.  Authors conclude, the study cannot conclude.

>>We agree with your point about attributing conclusions and findings correctly. We have revised the wording to indicate that the conclusions are made by the authors based on the study results.

4)  Please check to confirm the word count is within the manuscript parameters. 

>> we have ensured that the revised abstract conforms to the word limit set by the journal(200words)

“Here is our revised abstract:”

In response to rising environmental concerns and the increase in eco-friendly sports activities, this study investigated the determinants of sustained intention to participate in plogging, a combination of jogging and litter collection. A total of 288 randomly assigned plogging participants were surveyed to discern the effects of autonomy, competence, and relatedness experiences on sustained plogging intentions as suggested by Self-Determination Theory. The study also examined the moderating role of eco-friendly attitudes. The analysis, executed using multi-group structural equation modeling, revealed that while autonomy and competence did not significantly influence extrinsic motivation, relatedness emerged as the most influential factor. This suggests that plogging primarily serves as a prosocial behavior enhancing relationships, rather than a means to increase physical competence. The values derived from plogging and the intention to continue varied based on participants' eco-friendly attitudes. The authors conclude that voluntary participation and socialization are the core values of plogging and understanding these can promote healthier and more sustainable behaviors.

2. Introduction

1) In this context, many members of society have become more aware of the impact of their actions on the planet, leading to the rise of terms such as "environment", "eco friendly", "sustainability", and "green" as significant societal agendas [2].

= The intent of this sentence is understood, but I do not believe the grammar is correct. 

>>> Thank you for your feedback on the sentence. I apologize for any grammatical errors. I appreciate your attention to detail, and I hope the revised sentence accurately conveys the intended meaning while addressing any grammar concerns. If you have any further suggestions or feedback, please let me know.

Here is a revised version:

"In this context, the growing awareness among many members of society regarding the impact of their actions on the planet has resulted in the emergence of significant societal agendas, exemplified by terms such as 'environment,' 'eco-friendly,' 'sustainability,' and 'green' [2].“

2) Seek consistent in text citation format. 

>>In response to your feedback, we have checked and corrected the reference notation throughout the entire text. Thank you.

3) Backdrop? A lovely word, but I question is applicability here. 

>> Thank you for your valuable feedback regarding the use of the word "backdrop." I understand your concern about its appropriateness in this context. Here is a revised version:

"From this perspective, mature consumers are demonstrating a tendency towards 'pro-environmental behavior,' considering both environmental protection and consumption behavior [5].“

4) Line 39 and 42– avoid ‘This’. Be specific in your use of words. If a future author is to draw this sentence from this piece, the term ‘This’ can be construed. 

>> I appreciate your feedback. I apologize for the ambiguity caused by the use of the term "this." Here is a revised version that provides more clarity(specific use of words):

The tendency towards eco-friendly sports is also evident in sports consumption and participation behaviors [6]. Notable examples include the organization of eco-friendly sports events, the promotion of eco-friendly ideologies through sports, and the development and expansion of eco-friendly sports initiatives [6]. One such eco-friendly sport is plogging, which originated in Sweden and involves combining jogging with the act of picking up discarded litter. Plogging serves as a volunteer activity for environmental protection while also promoting individual health and well-being [7,8,9]. The engagement of individuals in these activities demonstrates their active participation in environmental preservation [7,8,9]. Moreover, these eco-friendly sports contribute to personal life satisfaction, the formation of self-identity, and overall happiness, as they simultaneously prioritize individual health and well-being along with environmental protection campaigns [10]. Previous research has highlighted the positive impact of participating in eco-friendly sports on personal life satisfaction, competence development, and physical fitness [7,8,9,11].

3. Theoretical Background and Research Hypotheses

1) Please be sure to have every pointed statement (a statement that expresses factual information) has a citation and accompanying reference. 

>> We have made efforts to enhance the references where they are needed.

2) Why is eco-friendly bolded? 

>> This was a mistake and it has been corrected. Thank you.

4. Research Method

1) Was IRB approval earned? 

>>This study was conducted in Korea, and I have gone through the IRB approval process through the institution I belong to.

2) Should the first table not be in the results section? 

>>his information corresponds to the data as demographic information of the respondents who participated in this study. Therefore, it is presented in the research methodology. However, if you prefer, we can move the demographic information of the respondents to the results section.

5. Results

1) Remove the brackets at the beginning of the paragraphs – [Table X].

>> Yes, all brackets have been removed from the main text.

2) Line 477 – If you have a First, you must have a Second. 

>> Following your comment, we have removed 'first' as it was not used to present parallel information.

6. Discussion and Conclusion

1) My overall recommendation for this section is to tighten up the language being used. 

 As difficult and ‘unromantic’ as it may be, being precise and concise with wording is the appropriate tact to take. State the facts. Reduce any lead-in terms (First, Additionally, Similarly).  Personally, I find this the most difficult aspect of writing manuscripts. I want to explain the research and convey what I believe to be exciting or interesting.

>> As you pointed out, we found parts that did not read as concisely as they should. We have also revised these throughout the text.

We have made every effort to revise the grammar and sentence expressions throughout. We have strived to diligently make corrections according to your comments and if there are any further deficiencies, we will certainly make additional revisions. Once again, we appreciate your meticulous review.

Reviewer 3 Report

The present research examined the relationship between Pro-environmental attitudes, self-determination, and sustained intention in eco-friendly sport participation. The main findings are indeed interesting with significant practical implementation for policy development to support eco friendly sports activities.

Author Response

We express our sincere gratitude for your thoughtful review of our manuscript, "Examining the Relationship between Pro-Environmental Attitudes, Self-Determination, and Sustained Intention in Eco-Friendly Sports Participation: A Study on Plogging Participants."

We are greatly encouraged by your affirmation of the study's significant findings and their potential practical implementation for policy development. Our intention was to provide a thorough investigation of the determinants of eco-friendly sports participation, and your recognition of the value of our research results is highly gratifying.

We look forward to our findings contributing positively to the development of policies that promote and support eco-friendly sports activities, and we thank you for your recognition of this potential impact.

Once again, we appreciate your constructive review and endorsement of our study. We remain committed to contributing high-quality research that advances understanding in this crucial area.

With sincere regards,

Round 2

Reviewer 2 Report

Well done with the revisions. 

Did you receive an approval number from your IRB. I truly respect your institutions IRB process, but also believe you should present some evidence (approval number) of the fact you sought clearance prior to engaging in the research process. 

I do believe the demographic information offered in Table 1 should be moved to the results section, as this information is the result of the research effort.

Other than that, I do not see any significant issues with the manuscript. 

Author Response

I hope this message finds you well.

Below are our responses in accordance with your suggestions.

1. “Did you receive an approval number from your IRB. I truly respect your institutions IRB process, but also believe you should present some evidence (approval number) of the fact you sought clearance prior to engaging in the research process. ”

>>>>

I'm writing to address your request regarding our Institutional Review Board (IRB) approval number.

Before initiating our study, we received an exemption review from our home institution, Dongguk University.

However, we are currently in a situation where we need to reconfirm the approval number for this exemption. Unfortunately, Dongguk University, our affiliated institution, is on shutdown until July 23rd.

Therefore, we will likely be able to complete the revision after confirming the IRB exemption approval number, at least after July 25rd.

Could you please let us know if this might pose a potential issue? If you could afford us a little more time, we will be able to submit the approval number for the exemption review.

2. I do believe the demographic information offered in Table 1 should be moved to the results section, as this information is the result of the research effort.

>>>>

Following your suggestion, we have moved the demographic information provided in Table 1 to the results section. Accordingly, we have made adjustments to the sequence of the tables and the corresponding text throughout the manuscript.

We greatly appreciate your thoughtful review.

Round 3

Reviewer 2 Report

Well done!

You have made significant improvements in the manuscript. I endorse this document for publication once very minor grammatical errors have been remedied. I believe there is a capitalization error at the beginning of the Results section. 

 I believe there is a capitalization error at the beginning of the Results section. 

Author Response

Thank you for your positive feedback and endorsement of the manuscript.

We appreciate your thorough review and identification of minor grammatical errors. We will carefully revise the manuscript to address the capitalization error at the beginning of the Results section. Y

our attention to detail is greatly valued, and we will ensure that the final version of the manuscript is error-free before submission for publication.